# SHAP-enhanced machine learning identifies modifiable obesity predictors across adolescent weight groups: A 2021 YRBSS analysis

Yuhai Peng[1], Zehan Xu[2], Songjian Du[3], Tianyuan Hou[3], Jin Yan[3]*

**1** School of Physical Education, Henan University of Economics and Law, Henan, China, **2** Faculty of Science, University of Sydney, Sydney, New South Wales, Australia, **3** School of Physical Education and Sports Science, Soochow University, Suzhou, China

* jinyan1126@suda.edu.cn

## Abstract

### Background

The growing prevalence of obesity in adolescents around the world poses a major threat to public health. This research uses machine learning models to examine the main causes of obesity, in contrast to standard information that typically rely on a single chance. The important fat-related steps were identified and ranked in this assessment to provide information on the effectiveness of the expected solutions.

### Methods

Data from the 2021 Youth Risk Behavior Surveillance System (YRBSS) were used in a cross-sectional analysis of adolescents aged 12–18 years. Random Forest and XGBoost models were implemented to investigate behavioral, dietary, sleep, and substance use factors. Model interpretability was enhanced using SHapley Additive exPlanations (SHAP).

### Results

Breakfast frequency, moderate-to-vigorous physical activity (MVPA) days, sleep duration, fruit intake, and screen time emerged as the most important predictors of obesity, with vaping also contributing to risk. Random Forest achieved an accuracy of 66.4% and XGBoost 66.3%, both with modest discriminative ability (AUC ~ 0.58). Fewer MVPA days, lower breakfast frequency, shorter sleep duration, lower fruit intake, and longer screen time were associated with increased obesity risk. SHAP analysis confirmed breakfast frequency and MVPA days as the top-ranked factors.

### Conclusion

Machine learning models identified key predictors of adolescent obesity, providing insights into the complex interplay of behavioral and lifestyle factors. Public health

**Data availability statement:** All relevant data are within the paper and its Supporting Information files.

**Funding:** The author(s) received no specific funding for this work.

**Competing interests:** NO authors have competing interests.

strategies should prioritize daily breakfast and fruit consumption, regular physical activity, sufficient sleep, reduced screen time, and vaping prevention to mitigate rising obesity rates among adolescents.

## 1 Introduction

The number of people with obesity has increased significantly around the world, affecting people of all ages [1]. Reports show that obesity around the world has nearly tripled since 1975. This increase is caused by biological, life, and social variables [2]. In 2016, over 650 million people were considered overweight, and the levels of obesity in children and adolescents were likewise increasing rapidly [3]. In the U.S., the number of overweight adolescents risen from 10.5% between 1988 and 1994 to 20.6% in 2017 and 2018, according to various national health surveys [4]. A study in 41 countries found that 8.9% of hungry adolescents were also overweight. This shows that in some regions, there are both thin and obese people at the same time [2].

In some areas of Asia, the increase in BMI is particularly significant in children and adolescents. In some rich countries, the rapid increase in BMI has slowed down and stopped growing as fast as before [2,4–6]. During adolescence, it is important to foster good behavior, but the growing rates of obesity during this time can lead to severe health problems later on. Research has shown that excessive weight gain during adolescence is associated to health problems in parents, like heart disease, type 2 diabetes, and some types of cancer [7–9]. Furthermore, being extremely heavy can harm mental health and the general quality of life. This makes adolescents especially sensitive to feelings like being judged by others and feeling sad or stressed in social situations [10]. Therefore, the objective of this study is to identify modifiable predictors of adolescent obesity using Shapley Additive exPlanations (SHAP)-enhanced machine learning models applied to the 2021 YRBSS dataset, in order to provide interpretable evidence for targeted public health interventions.

In the United States, many adolescents are suffering from obesity, and the latest statistics show that more than 20% of them are considered overweight. This matches information from national health surveys showing a significant rise in obesity rates among adolescents [11]. This design is part of a more extensive worldwide matter, with the number of obese children and adolescents rising nearly ten times over the past four decades [2,12]. The rising trend may be explained by lifestyle changes, such as less physical activity, unhealthy eating habits, more moments spent in front of screens, and quick access to bad foods that are high in calories but low in nutrients [13,14]. Moreover, family incomes and educational levels are significantly impacts whether individuals can get good food and opportunities to practice. This helps explain why some groups have higher rates of obesity than others [3]. YRBSS conducted biennially by the CDC since 1991, provides nationally representative data on health behaviors among U.S. high school students. The 2021 YRBSS dataset used in this study covers a wide range of demographic and behavioral factors relevant to adolescent obesity.

To handle adolescent obesity, we need to know how genes, activities, environment, and social variables all work together. This situation is made more complicated by cultural factors, city living, and new technologies that have caused people to be less active [15]. Obesity has many causes, so we need a careful way to prevent and treat it. This means concentrating on early prevention or primary prevention, creating specific health programs, and building a friendly environment that encourages healthy behavior [16,17].

Moreover, machine learning improves how effectively we can create and understand projections. Strategies like SHAP help us understand how much each data part affects the estimates. This makes understanding and applying machine learning designs for public health activities easier. For instance, SHAP has been successfully used to identify various health results and to know how different factors work together in intricate models [18,19]. This ability to focus on essential aspects allows for certain activities that are stronger at lowering the risk of obesity [20,21].

Even though it could be beneficial, applying machine learning in studies about obesity in adolescents is still not very popular. Most studies primarily focus on individual risk factors, rather than considering the broader context [22]. Structure understanding is still not used frequently in adolescent obesity information despite its potential benefits. Instead of focusing on the broader context, the majority of opinions are on special risk aspects. To capture potential differences across subpopulations, adolescents were also stratified by weight categories, which may reveal unique behavioral and environmental predictors of obesity. Utilizing machine learning models to identify and categorize the main factors of adolescent obesity, the aim of this study was to advance the field. In this study, influential predictors and their relative importance are revealed through the use of machine learning techniques [23]. This review provides a nuanced knowing that supports more precisely targeted public health interventions. In order to improve the transparency of the models, techniques like SHAP are employed; this allows for a more precise understanding of each factor's influence on the obesity prediction [18,24].

To determine the most important factors of adolescent obesity, this study analyzed machine learning models. Our method places emphasis on identifying potentiating aspects that could be addressed by comprehensible algorithms. The findings from this study are expected to enrich the current body of knowledge by delineating the major predictors of adolescent obesity, which can be leveraged to design targeted public health strategies [25]. By identifying the key indicators of adolescent obesity, public health practitioners may be better equipped to design and implement interventions aimed at reducing obesity rates among adolescents, thereby contributing to improved health outcomes and lower long-term healthcare costs [26,27].

## 2 Methods

### 2.1 Study design and participants

We used data from the 2021 Youth Risk Behavior Surveillance System (YRBSS), a nationally representative dataset that provides comprehensive information on adolescents' health behaviors in the United States for the cross-sectional design of this study. The YRBSS dataset was chosen because of its thorough analysis of important demographic, behavioral, and environmental factors [28,29]. The sample was representative of American children in grades 9–12 when the data were gathered using a stage-based cluster trial design [30]. The cross-sectional nature of the study made it possible to examine the connections between various elements and obesity status (accessed on 10 December 2024) [31].

In the review sample were young YRBSS study participants who were between the ages of 12 and 18 in 2021. A two-step imputation technique was applied to handle missing values and retain all available data. Specifically, demographic variables (age, sex) were used as auxiliary variables in SPSS to impute missing values, ensuring completeness for subsequent analyses. The initial results were based on the age (98 cases, 0.6%), sex, and median age (both at 15 years old) and SPSS mode, respectively. Stata used the ages and sexes as sign parameters to find any missing data by making a number of assumptions. This method made sure the data was trustworthy and accessible later for analysis [32,33].

Based on the YRBSS's de-identified, publicly available information, the best administrative review panel gave approval to the research. Ethical approvals for the study were secured at the national or regional level, with each country obtaining approval from an ethics review board or an equivalent regulatory body specific to the government. The current study was approved by the Ethics Committee at Soochow University (SUDA20240626H06). In conducting this study, we adhered to the guidelines outlined in the Strengthening the Reporting of Observational Studies in Epidemiology (STROBE) [34].

## 2.2 Measurement of variables

Weight status was defined using the CDC sex-specific BMI-for-age percentiles (BMIPCT variable in YRBSS). Participants were classified as underweight (< 5th percentile), normal weight (5th – 84th percentile), overweight (85th – 94th percentile), or obese (≥ 95th percentile) [35]. The YRBSS evaluation, which had a variety of options for getting socioeconomic, cognitive, and health-related data, was used to analyze the factors in this review. Interviewees' ages ranged from "12 years old or younger" to "18 years older or older" when asked how old they were. The gender was chosen, regardless of whether it was "Male" or "Female". The two-step study asked respondents to be "White", "Black or African American", "Asian", "Hispanic/Latino" and others. For analysis, these responses were consolidated into four main categories: White, Black or African American, Hispanic/Latino, and All Other Races.

Moderate-to-vigorous physical activity (MVPA) days was measured by asking how many days in the past week participants were physically active for at least 60 minutes, with responses ranging from "0 days" to "7 days." Screen time was assessed based on the number of hours spent in front of screens on an average school day, excluding schoolwork, with options ranging from "less than 1 hour per day" to "5 or more hours per day." Sleep duration was determined by asking how many hours participants typically slept on school nights, with options from "4 or less hours" to "10 or more hours."

Dietary habits were captured through questions regarding the frequency of eating breakfast, consuming fruit juice, fruit, vegetables (green salad, potatoes, carrots, and other vegetables), and drinking soda and milk in the past week, with responses indicating frequencies from "0 days" or "I did not consume" to "7 days" or "4 or more times per day." Substance use was measured for alcohol, cigarettes, electronic vapor products, marijuana, and prescription pain medicine misuse. Participants were asked how many days in the past 30 days they used alcohol, cigarettes, or electronic vapor products, with response options ranging from "0 days" to "All 30 days." For marijuana and pain medicine misuse, participants were asked how many times in their lifetime they had used these substances, with answers ranging from "0 times" to "100 or more times."

## 2.3 Statistical analysis

In this study, we applied both conventional statistical methods and machine learning models to examine factors associated with adolescent obesity. The 2021 YRBSS, which contained a variety of health behavior data, was used as the analytic dataset. To decide which of the 20 potential indicators should be used, Chi-square tests were applied to screen candidate variables. Variables showing statistical significance ($p < 0.05$) in Chi-square tests were retained, resulting in 16 features that included demographic, behavioral, and environmental factors relevant to adolescent obesity [36]. For these statistical analyses, R software (version 4.4.0) was used [28].

To complement the regression analyses, we implemented two ML algorithms—Random Forest and XGBoost—to provide additional insights into variable importance in complex datasets [37]. Training (70%) and testing (30%) sets were included in the dataset. A fixed random seed (set.seed = 3554) was applied to ensure reproducibility. Hyperparameter tuning was performed using grid search with cross-validation, focusing on key parameters of each model. To address class imbalance, Synthetic Minority Over-sampling Technique (SMOTE) was applied as the primary method, supplemented by class weight adjustment. Model performance was evaluated using repeated 5-fold cross-validation on the training set and then assessed on the held-out testing set (30%) using accuracy, sensitivity, specificity, AUC of the ROC curve, and weighted F1-score [38]. Classification thresholds were set according to the default behavior of the algorithms, where each observation was assigned to the class with the highest predicted probability (argmax rule). In addition, imbalance-sensitive

metrics were reported, including the mean one-vs-rest AUC, which are particularly relevant given the unequal distribution of weight categories in the YRBSS dataset. These evaluations were used primarily to characterise model behavior rather than to optimise predictive accuracy.

The primary correlates of overweight and obesity were examined using multivariable logistic regression. Odds ratios (ORs) and 95% confidence intervals (CIs) were used to calculate statistical significance when the p-value was less than 0.05. For this analysis, SPSS (version 27.0) was used.

To further interpret model performance, we examined the relationships between predictors and outcomes. Feature importance was assessed using the Mean Decrease Gini index in the Random Forest model, whereas Gain quantified each feature's contribution to node splits in the XGBoost model [39]. SHAP were implemented, following Han and Wang (2023) [39]. For each model, SHAP values were computed for individual predictions and then aggregated across all participants using mean absolute SHAP values to determine overall feature importance. The distribution of SHAP values for each predictor was visualised using beeswarm plots, and the top 15 predictors of adolescent obesity were highlighted. Analyses and visualisations were conducted using R (version 4.4.0) and Origin (version 2021). Compared with the initial submission, the revised analyses incorporated hyperparameter tuning, imbalance handling, and cross-validation to strengthen robustness.

## 3 Results

Table 1 presents the distribution of demographic and behavioral characteristics across weight categories. Significant associations were observed for sex, age, and race/ethnicity. Obesity prevalence was higher among males (60.2%) than females (39.8%) ($p = 0.022$), and older adolescents (16–18 years) had a higher prevalence of obesity ($p < 0.001$). Obesity prevalence was highest among Hispanic/Latino participants (24.1%); the proportion of White participants within the obese group was 45.7% ($p < 0.001$). More detailed information is provided in the supplementary materials.

Table 2 provides the multivariable logistic regression results. Male participants were more likely to be overweight or obese compared to females (OR = 1.30, 95% CI: 1.22–1.40, $p < 0.001$). Race/ethnicity also played a substantial role: Black

**Table 1. Demographic characteristics across different weight categories.**

| Variables | Overall (n = 17232) | Non-overweight (n = 11459) | Overweight (n = 2749) | Obese (n = 3024) | $\chi^2$ | $p$ |
|---|---|---|---|---|---|---|
| | n (%) | n (%) | n (%) | n (%) | | |
| **Sex** | | | | | | |
| Female | 8152(47.3) | 5564(48.6) | 1384(50.3) | 1204(39.8) | 23.684 | 0.022 |
| Male | 9080(52.7) | 5895(51.4) | 1365(49.7) | 1820(60.2) | | |
| **Age** | | | | | | |
| 12 years old | 39(0.2) | 18(0.2) | 10(0.4) | 11(0.4) | 85.446 | < 0.001 |
| 13 years old | 62(0.4) | 38(0.3) | 13(0.5) | 11(0.4) | | |
| 14 years old | 3403(19.7) | 2227(19.4) | 601(21.9) | 575(19.0) | | |
| 15 years old | 4525(26.3) | 2997(26.2) | 728(26.5) | 800(26.5) | | |
| 16 years old | 4276(24.8) | 2857(24.9) | 635(23.1) | 784(25.9) | | |
| 17 years old | 3904(22.7) | 2638(23.0) | 607(22.1) | 659(21.8) | | |
| 18 years old | 1023(5.9) | 684(6.0) | 155(5.6) | 184(6.1) | | |
| **Race** | | | | | | |
| White | 9375(54.4) | 6641(58.0) | 1351(49.1) | 1383(45.7) | 257.976 | < 0.001 |
| Black or African American | 2384(13.8) | 1401(12.2) | 434(15.8) | 549(18.2) | | |
| Hispanic/Latino | 3331(19.3) | 1943(17.0) | 659(24.0) | 729(24.1) | | |
| All Other Races | 2142(12.4) | 1474(12.9) | 305(11.1) | 363(12.0) | | |

**Table 2. Multivariable logistic regression analysis of factors associated with overweight and obesity.**

| Variables | Odds Ratio | 95%CI | p |
|---|---|---|---|
| **Age** *(ref = 12 years old)* | | | |
| 13 years old | 0.76 | 0.33-1.76 | 0.522 |
| 14 years old | 0.70 | 0.36-1.35 | 0.292 |
| 15 years old | 0.67 | 0.35-1.29 | 0.228 |
| 16 years old | 0.64 | 0.33-1.24 | 0.187 |
| 17 years old | 0.61 | 0.32-1.17 | 0.139 |
| 18 years old | 0.63 | 0.32-1.22 | 0.172 |
| **Sex** *(ref = Female)* | | | |
| Male | 1.30 | 1.22-1.40 | < 0.001 |
| **Race** *(ref = White)* | | | |
| Black or African American | 1.51 | 1.37-1.67 | < 0.001 |
| Hispanic/Latino | 1.58 | 1.45-1.72 | < 0.001 |
| All Other Races | 1.02 | 0.92-1.13 | 0.683 |
| **MVPA days** *(ref = 0 days)* | | | |
| 1 day | 0.95 | 0.83-1.10 | 0.512 |
| 2 days | 0.95 | 0.84-1.08 | 0.452 |
| 3 days | 0.88 | 0.78-0.99 | 0.049 |
| 4 days | 0.77 | 0.68-0.88 | < 0.001 |
| 5 days | 0.84 | 0.75-0.95 | 0.007 |
| 6 days | 0.55 | 0.47-0.65 | < 0.001 |
| 7 days | 0.63 | 0.56-0.70 | < 0.001 |
| **Screen time** *(ref = Less than 1 hour per day)* | | | |
| 1 hour per day | 1.00 | 0.82-1.22 | 0.994 |
| 2 hours per day | 1.05 | 0.90-1.24 | 0.527 |
| 3 hours per day | 0.94 | 0.80-1.09 | 0.416 |
| 4 hours per day | 1.01 | 0.87-1.18 | 0.861 |
| 5 hours per day | 1.12 | 0.97-1.29 | 0.112 |
| **Sleep duration** *(ref = 4 or less hours)* | | | |
| 5 hours | 0.98 | 0.86-1.12 | 0.794 |
| 6 hours | 0.98 | 0.87-1.10 | 0.706 |
| 7 hours | 0.89 | 0.79-1.00 | 0.056 |
| 8 hours | 0.98 | 0.86-1.12 | 0.785 |
| 9 hours | 0.99 | 0.81-1.20 | 0.886 |
| 10 or more hours | 1.04 | 0.80-1.35 | 0.785 |
| **Breakfast days** *(ref = 0 days)* | | | |
| 1 day | 0.87 | 0.77-0.98 | 0.023 |
| 2 days | 0.94 | 0.84-1.05 | 0.270 |
| 3 days | 0.73 | 0.64-0.82 | < 0.001 |
| 4 days | 0.74 | 0.64-0.85 | < 0.001 |
| 5 days | 0.76 | 0.66-0.87 | < 0.001 |
| 6 days | 0.65 | 0.55-0.77 | < 0.001 |
| 7 days | 0.56 | 0.51-0.63 | < 0.001 |
| **Fruit** *(ref = Did not eat)* | | | |
| 1 to 3 times | 1.06 | 0.95-1.18 | 0.283 |
| 4 to 6 times | 1.15 | 1.02-1.30 | 0.021 |
| 1 time per day | 1.00 | 0.87-1.16 | 0.977 |

*(Continued)*

**Table 2.** (Continued)

| Variables | Odds Ratio | 95%CI | p |
|---|---|---|---|
| 2 times per day | 1.05 | 0.91-1.21 | 0.529 |
| 3 times per day | 1.18 | 0.98-1.42 | 0.075 |
| 4 or more times per day | 1.18 | 0.98-1.41 | 0.076 |
| **Other vegetables** *(ref=Did not eat)* | | | |
| 1 to 3 times | 1.01 | 0.92-1.11 | 0.856 |
| 4 to 6 times | 1.12 | 1.01-1.24 | 0.040 |
| 1 time per day | 1.01 | 0.88-1.16 | 0.838 |
| 2 times per day | 1.06 | 0.90-1.25 | 0.475 |
| 3 times per day | 1.32 | 1.06-1.65 | 0.015 |
| 4 or more times per day | 1.07 | 0.84-1.37 | 0.582 |
| **Soda** *(ref=Did not drink)* | | | |
| 1 to 3 times | 1.01 | 0.93-1.10 | 0.754 |
| 4 to 6 times | 1.04 | 0.94-1.16 | 0.459 |
| 1 time per day | 1.01 | 0.87-1.16 | 0.925 |
| 2 times per day | 1.05 | 0.89-1.23 | 0.591 |
| 3 times per day | 1.18 | 0.93-1.49 | 0.181 |
| 4 or more times per day | 1.18 | 0.96-1.44 | 0.121 |
| **Milk** *(ref=Did not drink)* | | | |
| 1 to 3 times | 1.06 | 0.97-1.15 | 0.177 |
| 4 to 6 times | 1.14 | 1.02-1.28 | 0.021 |
| 1 time per day | 1.11 | 0.99-1.24 | 0.075 |
| 2 times per day | 0.99 | 0.86-1.14 | 0.886 |
| 3 times per day | 1.10 | 0.89-1.35 | 0.376 |
| 4 or more times per day | 1.03 | 0.85-1.26 | 0.755 |
| **Alcohol** *(ref=0 days)* | | | |
| 1 or 2 days | 0.88 | 0.79-0.99 | 0.026 |
| 3 to 5 days | 0.85 | 0.73-1.00 | 0.051 |
| 6 to 9 days | 0.73 | 0.58-0.91 | 0.005 |
| 10 to 19 days | 0.93 | 0.70-1.23 | 0.599 |
| 20 to 29 days | 1.24 | 0.72-2.15 | 0.435 |
| All 30 days | 1.12 | 0.69-1.82 | 0.656 |
| **Cigarettes** *(ref=0 days)* | | | |
| 1 or 2 days | 1.53 | 1.20-1.95 | < 0.001 |
| 3 to 5 days | 0.96 | 0.63-1.44 | 0.834 |
| 6 to 9 days | 1.17 | 0.64-2.15 | 0.604 |
| 10 to 19 days | 1.49 | 0.92-2.41 | 0.102 |
| 20 to 29 days | 1.11 | 0.49-2.52 | 0.803 |
| All 30 days | 0.95 | 0.60-1.49 | 0.817 |
| **Electronic vapor** *(ref=0 days)* | | | |
| 1 or 2 days | 0.94 | 0.79-1.11 | 0.454 |
| 3 to 5 days | 1.24 | 1.01-1.53 | 0.045 |
| 6 to 9 days | 1.07 | 0.83-1.38 | 0.612 |
| 10 to 19 days | 0.95 | 0.76-1.19 | 0.682 |
| 20 to 29 days | 1.01 | 0.79-1.29 | 0.936 |
| All 30 days | 1.07 | 0.89-1.27 | 0.488 |

*(Continued)*

**Table 2.** (Continued)

| Variables | Odds Ratio | 95%CI | p |
|---|---|---|---|
| **Marijuana** *(ref = 0 times)* | | | |
| 1 or 2 times | 1.09 | 0.94-1.28 | 0.253 |
| 3 to 9 times | 1.12 | 0.92-1.36 | 0.251 |
| 10 to 19 times | 1.12 | 0.89-1.40 | 0.352 |
| 20 to 39 times | 0.85 | 0.66-1.10 | 0.225 |
| 40 or more times | 0.86 | 0.70-1.06 | 0.149 |
| **Pain medicine** *(ref = 0 times)* | | | |
| 1 or 2 times | 1.11 | 0.92-1.33 | 0.280 |
| 3 to 9 times | 1.01 | 0.78-1.30 | 0.929 |
| 10 to 19 times | 1.10 | 0.75-1.62 | 0.631 |
| 20 to 39 times | 1.05 | 0.54-2.04 | 0.877 |
| 40 or more times | 1.19 | 0.75-1.89 | 0.460 |

or African American (OR = 1.51, 95% CI: 1.37–1.67, *p*<0.001) and Hispanic/Latino adolescents (OR = 1.58, 95% CI: 1.45–1.72, *p*<0.001) had higher odds of obesity than White participants. MVPA on 6 days per week (OR = 0.55, 95% CI: 0.47–0.65, *p*<0.001) and daily breakfast consumption (OR = 0.56, 95% CI: 0.51–0.63, *p*<0.001) showed the strongest protective effects. For screen time, adolescents reporting ≥5 hours per day had higher odds of obesity (OR = 1.12, 95% CI: 0.97–1.29, *p*=0.112) compared with those reporting less than 1 hour per day.

Table 3 compares the performance of the Random Forest and XGBoost models. The Random Forest model achieved a slightly higher accuracy (66.4%) than XGBoost (66.3%), with very similar AUC values (0.577 vs. 0.576). Both models demonstrated extremely low specificity (3.9% vs. 4.8%), indicating limited ability to correctly identify negative cases. Random Forest showed a higher negative predictive value (0.60 vs. 0.47), whereas XGBoost performed marginally better on positive predictive value (0.677 vs. 0.672). The F1 scores were comparable (0.800 vs. 0.797), suggesting balanced performance on precision and recall for the positive class.

Fig 1 shows the feature importance in both models based on the Mean Decrease Gini index (Random Forest) and Gain metrics (XGBoost). "Breakfast days" and "MVPA days" consistently emerged as the top predictors across both models, followed by "Sleep duration" and "Fruit" intake. The highest Mean Decrease Gini index was observed for Breakfast days, highlighting its crucial role in predicting obesity.

Fig 2 displays the beeswarm plot of SHAP values, highlighting the top 15 predictors of adolescent obesity in the XGBoost model. Higher MVPA days, longer sleep duration, and more frequent breakfast consumption were associated

**Table 3. Comparison of evaluation performance between random forest and XGBoost.**

| | Random Forest | XGBoost |
|---|---|---|
| Accuracy | 0.6636 | 0.6625 |
| 95% CI | (0.6506, 0.6765) | (0.6494, 0.6754) |
| Mcnemar's Test P-Value | < 0.001 | < 0.001 |
| Specificity | 0.0389 | 0.0477 |
| Pos Pred Value | 0.6720 | 0.6765 |
| Neg Pred Value | 0.6000 | 0.4655 |
| F1 | 0.7995 | 0.7972 |
| AUC | 0.5769 | 0.5763 |

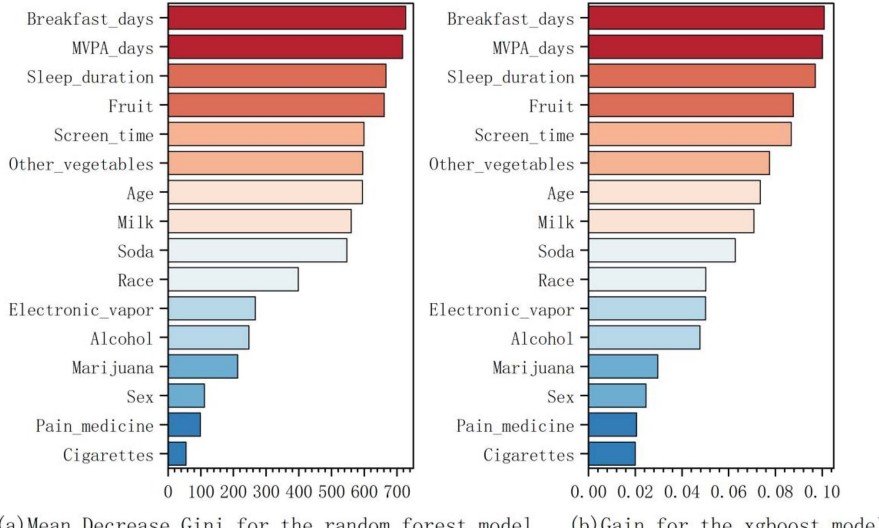

(a) Mean Decrease Gini for the random forest model      (b) Gain for the xgboost model

**Fig 1. Importance ranking of factors influencing youth obesity with Random Forest and XGBoost models.**

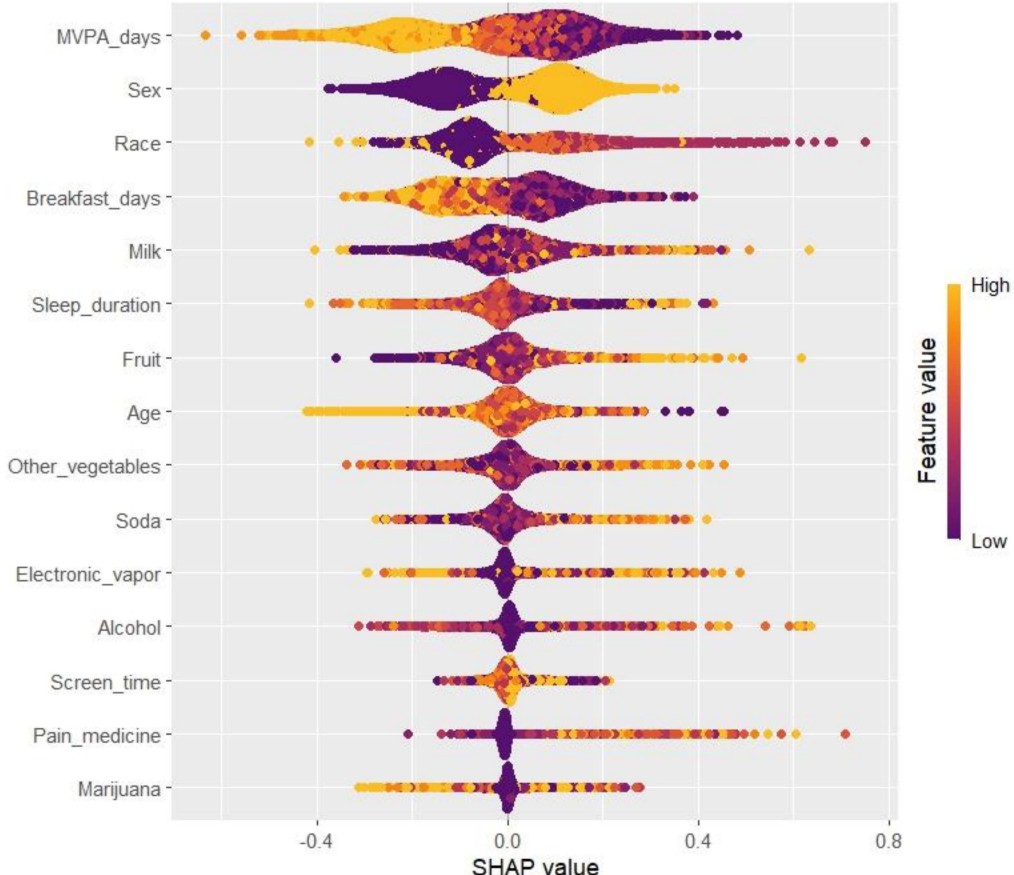

**Fig 2. Beeswarm plot of SHAP values for the top 15 predictors of adolescent obesity in the XGBoost model.** (Note: Each point represents an individual participant. The x-axis shows the SHAP value (impact on log-odds of obesity), where positive values indicate increased risk and negative

values indicate reduced risk. The y-axis lists predictors ranked by overall importance, and point colour reflects the feature value (yellow=high, purple=low). The least influential variable (Cigarettes) was excluded).

with reduced obesity risk, whereas higher screen time, greater soda intake, and electronic vapor use contributed to increased risk. The top features were ranked according to their mean absolute SHAP values across all participants, with point colours indicating feature values (yellow=high, purple=low).

Fig 3 presents a nomogram constructed based on the Random Forest and XGBoost models, illustrating the relative contributions of key predictors to obesity risk. The nomogram assigns point values to predictors such as age, race, MVPA days, and dietary habits. Higher total scores, particularly those associated with fewer MVPA days, shorter sleep duration, lower breakfast frequency, and increased screen time, indicated a higher likelihood of obesity.

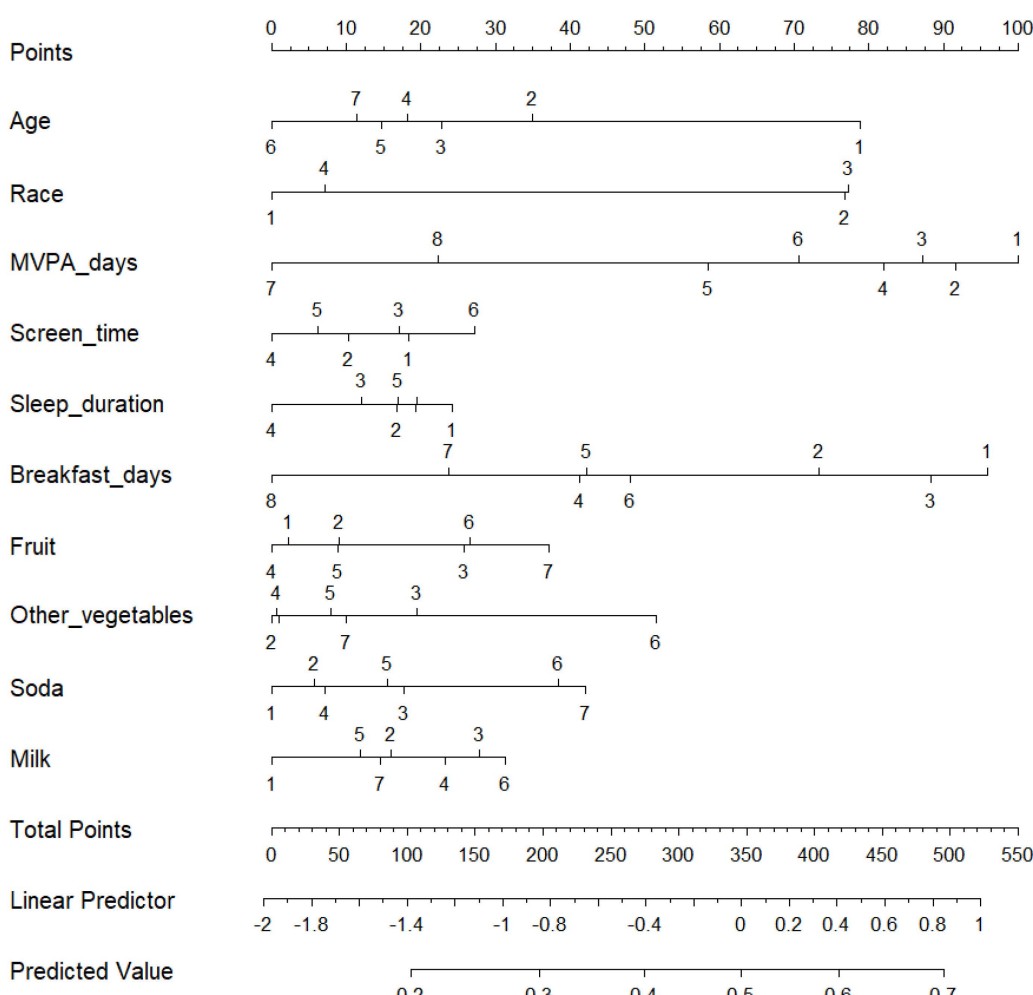

**Fig 3. Nomogram of predictors of adolescent obesity.** (Note: Each predictor is shown as a horizontal scale, with values mapped to points according to their relative contribution. The "Points" axis at the top indicates the score assigned to each predictor level. Summing all scores yields a "Total Points" value, which is then projected onto the "Linear Predictor" and "Predicted Value" axes at the bottom to estimate the probability of obesity. Higher total points correspond to greater predicted obesity risk).

## 4  Discussion

This study applied machine learning models (Random Forest and XGBoost) combined with SHAP to identify key predictors of adolescent obesity. The use of SHAP improved model interpretability by quantifying the contribution of each predictor to obesity risk, thereby bridging the gap between complex modelling and clinically relevant insights [40,41]. Both machine learning and traditional regression analyses consistently highlighted breakfast frequency, MVPA days, sleep duration, fruit intake, and screen time as the most important modifiable predictors of obesity [42–44]. These findings strengthen the evidence base for prioritising behavioral interventions targeting these factors in adolescent obesity prevention.

### MVPA

In this study, MVPA days were identified as one of the strongest predictors of an adolescent's BMI status, ranking just after breakfast frequency. According to research that links physical activity to maintaining normal weight, adolescents who regularly participated in MVPA were significantly less likely to be overweight or obese [45,46]. Increased muscle mass, improved insulin sensitivity, and increasing energy expenditure, all of which contribute to lowering natural BMI increases, may be a result of MVPA's efforts to lower fat regulations. Since then, Wang et. al emphasize the value of regular physical activity, who found that adolescents who adhered to the daily MVPA guidelines were significantly less likely to become obese [47]. Similarly, Lister et al. (2023) demonstrated that maintaining physical activity during adolescence is essential to preventing obesity later in life [48]. In addition to the WHO guidelines recommending at least 60 minutes of MVPA per day, these findings support the necessity of school-based activities that use normal physical activity for all children. In both Random Forest and XGBoost models, MVPA days consistently ranked among the top predictors, second only to breakfast frequency. This finding matches the data Klein et al. provided, which demonstrated that school initiatives that promote regular physical activity lower obesity prevalence and improve long-term health outcomes [49]. Given the significant effect of physical activity on BMI, increasing options for MVPA should be a crucial focus of public health approaches aimed at adolescent obesity prevention.

### Screen time

Although screen time emerged as a relevant predictor in ML models, its association with obesity was not statistically significant in regression analyses ($p = 0.112$). This discrepancy highlights the complexity of behavioral influences and the value of complementary analytic approaches. Adolescents reporting ≥5 hours of daily screen time had slightly higher odds of obesity compared with those reporting <1 hour, although this association did not reach statistical significance. This objective is in line with the findings of Christofaro et al. (2016) and Al-Hazzaa (2018), which established a clear link between excessive screen time and poor eating habits [50,51]. One possible pathway linking prolonged screen exposure to higher BMI is through displacement of physical activity and promotion of obesogenic behaviors such as snacking. Buchanan et al. (2016) made a case for the benefits of sedentary behavior and how screen media exposure can lead to weight gain [52]. Public health interventions should therefore aim to reduce excessive screen exposure and encourage more active alternatives to limit its potential impact on BMI.

### Dietary predictors

Breakfast frequency was identified as the strongest dietary factors associated with a lower risk of high BMI. According to previous research, adolescents who had breakfast daily were significantly less overweight or obese [53,54]. According to Szajewska & Ruszczynski (2010), better appetite regulation and a healthier overall diet are directly related to one's BMI [55]. Additionally, Gordon-Larsen et al. (2006) found who have breakfast are more likely to maintain a balanced diet throughout the day to prevent overeating or consuming unhealthy snacks [56]. Adolescents who frequently skip breakfast

are more likely to have a higher BMI as a result of changes in their daily eating patterns [57]. Our regression and machine learning results consistently demonstrated that daily breakfast consumption was strongly protective against obesity, aligning with Al-Hazzaa et al.(2012), which emphasizes the importance of breakfast in adolescent health [58].

Milk intake also appeared as a secondary dietary factor associated with adolescent BMI. Regression findings were inconsistent across intake levels (including an elevated odds ratio at 4–6 times/week), suggesting milk's association with BMI may depend on type and context of consumption. Milk provides essential nutrients such as calcium, protein, and vitamin D, which are important for metabolic health and bone development [59]. However, the relationship with BMI may vary by type (whole vs. low-fat) and frequency of consumption. While our study only measured overall milk frequency, future research should differentiate by milk type to refine dietary recommendations [60–62]. Fruit intake emerged as an important predictor in the ML/SHAP analysis; however, regression results were mixed across consumption categories, with one intermediate level associated with higher odds of obesity. This discrepancy may reflect measurement and residual confounding. Adolescents reporting higher fruit consumption were less likely to be obese, consistent with evidence that fruit contributes to satiety and healthier dietary patterns [63,64]. In contrast, vegetable consumption showed less consistent associations, possibly due to differences in measurement and preparation methods [65]. These findings suggest that promoting daily fruit intake, alongside other healthy dietary practices, may play a meaningful role in reducing obesity risk.

## Substance Use

Electronic vapor use was associated with higher BMI in the machine learning models, whereas cigarette use did not consistently emerge as a significant predictor. This relationship may be driven by behavioral patterns such as reduced physical activity and poor dietary choices, which often cluster with substance use [66,67]. Addressing vaping in adolescence may therefore have additional benefits for obesity prevention alongside its well-established health risks [68]. The associations between alcohol and marijuana use with BMI were less clear, as these substances did not consistently predict obesity risk across analyses [69,70]. The unexpected inverse association with alcohol in some models likely reflects residual confounding related to adolescent social behaviors rather than a direct effect of alcohol consumption. Future studies should explore these relationships in greater depth, considering other behavioral and environmental influences that may contribute to the observed trends [71].

## Implications

The need for qualified public health interventions focused on these crucial areas is underscored by the more substantial effects of physical activity, breakfast frequency, sleep duration, and fruit intake on BMI, with screen time playing a more modest role. Schools should establish rules to reduce excessive screen exposure, encourage healthier eating habits, and increase opportunities for daily physical activity. To avoid negative impacts on BMI, it is also crucial to address adolescent substance use, particularly vaping/e-cigarette use. Public health professionals can use the insights from machine learning models to inform interventions, while recognising the modest predictive performance of these models [72,73].

## Study limitations

Although this research contains important data, there are some limitations. First, the cross-sectional design prevents causal inference, and the identified associations should be interpreted as correlational rather than causal. Second, all measures relied on self-reported data, which introduces the potential for recall and reporting bias [74], particularly in physical activity, nutrition, and substance use. Third, we did not include longitudinal validation, so it remains uncertain whether the identified predictors can consistently forecast obesity risk over time. Additionally, model performance was modest (accuracy ~66%, with very low specificity), reflecting reliance on self-reported behavioral data. Despite tuning and imbalance handling, predictive power remained limited; thus, the ML analyses were intended to highlight correlates rather

than provide precise prediction. Future research should therefore adopt longitudinal designs, incorporate wearable-based measures, and expand the scope of machine learning models to include genetic and environmental factors, which could further improve predictive power and utility in public health applications [75,76].

## 5 Conclusion

This study provides valuable insight into the most critical predictors of adolescent obesity, particularly breakfast frequency, MVPA, sleep duration, fruit intake, screen time, and vaping. It also demonstrates how machine learning models can complement traditional regression by highlighting key behavioral and dietary factors, despite their modest predictive performance. By targeting these behaviors, public health professionals can develop more effective interventions to reduce adolescent obesity and improve long-term health outcomes.

## Supporting information

**S1. 2021 YRBS Data Working V1 10.1.**
(XLSX)

## Acknowledgments

The authors sincerely appreciate all the students who participated in this study, especially the dedicated fieldworkers, for their invaluable contributions to data collection.

## Author contributions

**Conceptualization:** Yuhai Peng, Songjian Du.

**Data curation:** Yuhai Peng.

**Formal analysis:** Yuhai Peng.

**Methodology:** Zehan Xu.

**Project administration:** Zehan Xu.

**Resources:** Zehan Xu, Tianyuan Hou.

**Software:** Zehan Xu, Tianyuan Hou.

**Supervision:** Tianyuan Hou, Jin Yan.

**Validation:** Songjian Du, Tianyuan Hou, Jin Yan.

**Visualization:** Songjian Du, Tianyuan Hou, Jin Yan.

**Writing – original draft:** Songjian Du, Jin Yan.

**Writing – review & editing:** Jin Yan.

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
