## [Decision Letter · Decision Letter 0]

28 Jul 2025

PONE-D-25-31368Machine Learning Analysis of Obesity Factors Across Weight Groups: Findings from the 2021 Youth Risk Behavior Surveillance SystemPLOS ONE

Dear Dr. Yan,

Thank you for submitting your manuscript to PLOS ONE. After careful consideration, we feel that it has merit but does not fully meet PLOS ONE’s publication criteria as it currently stands. Therefore, we invite you to submit a revised version of the manuscript that addresses the points raised during the review process.

Please submit your revised manuscript by Sep 11 2025 11:59PM. If you will need more time than this to complete your revisions, please reply to this message or contact the journal office at plosone@plos.org . Please include the following items when submitting your revised manuscript:

We look forward to receiving your revised manuscript.

Kind regards,

Mohamed Ahmed Said, Ph.D.

Academic Editor

PLOS ONE

5. Please amend your authorship list in your manuscript file to include author Songjian Du ,

6. Please include a separate caption for each figure in your manuscript.

Additional Editor Comments:

Manuscript Title:

Machine Learning Analysis of Obesity Factors Across Weight Groups: Findings from the 2021 Youth Risk Behavior Surveillance System

General Assessment:

This manuscript addresses a timely and relevant public health issue—adolescent obesity—by applying supervised machine learning models to a large national dataset (YRBSS 2021). The incorporation of SHAP values to interpret feature importance is a methodological strength, offering transparency and clinical relevance. However, the manuscript requires substantial revision to clarify methodological choices, strengthen the scientific communication of results, and improve reproducibility and interpretability.

1. Title and Introduction

Concerns:

• The title does not reflect the use of SHAP or the study's focus on modifiable predictors, which are core to its novelty.

• The study aim appears too late in the introduction.

• The YRBSS dataset is not introduced early or described sufficiently.

• There is no justification provided for stratifying by weight categories.

Recommendations:

• Revise the title to reflect key methodological features. Suggested title:

“SHAP-Enhanced Machine Learning Identifies Modifiable Obesity Predictors Across Adolescent Weight Groups: A 2021 YRBSS Analysis.”

• State the study objective clearly and early.

• Briefly describe the YRBSS dataset (e.g., scope, periodicity, population coverage).

• Justify the weight-group stratification using appropriate references (e.g., CDC percentile cutoffs for BMI-for-age).

2. Methods

Concerns:

• Unclear terminology (e.g., “two-step say technique”) may confuse readers.

• The definition of obesity and BMI percentile cutoffs is missing.

• Critical aspects of the machine learning pipeline are underreported:

o No mention of hyperparameter tuning

o No treatment of class imbalance

o No description of cross-validation

• SHAP implementation details are insufficient.

• Reproducibility elements (software, random seed, code repository) are lacking.

Recommendations:

• Correct unclear terms or provide definitions.

• Specify how BMI categories were defined and cite relevant standards (e.g., CDC, WHO).

• Expand the ML pipeline description to include:

o Hyperparameter optimization approach

o Class imbalance techniques (e.g., SMOTE, weighting)

o Validation strategy (e.g., k-fold CV, train-test split)

• Clearly explain how SHAP values were computed and aggregated (e.g., mean absolute values).

• For reproducibility, list the software used, indicate random seed values, and share a code repository if available.

3. Results

Concerns:

• The results section includes redundant reporting (e.g., odds ratios repeated in text and tables).

• Overreporting of non-significant findings (e.g., soda, sleep) may distract from meaningful results.

• Performance metrics sensitive to class imbalance (e.g., F1-score, PR AUC) are missing.

• SHAP plots and nomograms are insufficiently labeled.

Recommendations:

• Streamline the text to highlight key, statistically significant findings.

• Move non-significant results to supplementary material.

• Report additional evaluation metrics (e.g., F1-score: “F1 scores were 0.41 for Random Forest and 0.39 for XGBoost”).

• Improve figure labeling:

o Label SHAP axes (e.g., “SHAP value (impact on log-odds of obesity)”)

o Clarify nomogram elements with legends or captions.

• State that top features were selected based on mean absolute SHAP values.

4. Discussion

Concerns:

• The discussion contains redundant commentary on interpretability.

• Terminology is inconsistent or unclear (e.g., “butter level,” “fresh BMI,” “quiet activities”).

• Overgeneralizations are made, particularly regarding dietary predictors.

• The narrative flow is disjointed.

• The limitations section is underdeveloped.

Recommendations:

• Consolidate discussion on SHAP and interpretability.

• Use consistent, scientific terminology (e.g., “butter level” → “milk intake,” “fresh BMI” → “BMI status”).

• Clarify overgeneralized findings (e.g., specify milk type/frequency).

• Improve discussion structure using subheadings (e.g., MVPA, screen time, substance use).

• Expand the limitations section:

o Cross-sectional design (no causal inference)

o Self-reported data bias

o Lack of objective activity measures (e.g., wearables)

o Absence of longitudinal validation

5. Statistical Analysis Clarity

Concern:

The authors mention the use of accuracy, sensitivity, specificity, and AUC to evaluate model performance but do not detail how these metrics were calculated or validated.

Recommendation:

• Briefly describe whether performance metrics were calculated on a holdout set or through cross-validation.

• Indicate how classification thresholds were set.

• Discuss the relevance of class imbalance metrics if applicable.

6. Minor Issues

• Proofread for minor typographical and grammatical inconsistencies.

• Standardize variable names and phrasing throughout the manuscript.

Final Recommendation:

Major Revision

The manuscript presents a valuable application of machine learning to a pressing health issue, but critical revisions are necessary to ensure scientific clarity, methodological transparency, and interpretability. I encourage the authors to address the above comments comprehensively.

Reviewers' comments:

Reviewer's Responses to Questions

**Comments to the Author**

1. Is the manuscript technically sound, and do the data support the conclusions?

Reviewer #1: No

Reviewer #2: Yes

2. Has the statistical analysis been performed appropriately and rigorously? 

Reviewer #1: I Don't Know

Reviewer #2: Yes

3. Have the authors made all data underlying the findings in their manuscript fully available?

Reviewer #1: Yes

Reviewer #2: Yes

4. Is the manuscript presented in an intelligible fashion and written in standard English?

Reviewer #1: No

Reviewer #2: Yes

5. Review Comments to the Author

Reviewer #1: Thank you for the opportunity to review your manuscript. This study analyzes a large dataset, which is a valuable strength and holds great potential to contribute to the field. I appreciate the effort that went into collecting and analyzing the data.

That said, there are a few areas where the manuscript could be improved to better communicate your findings and enhance the overall impact of the paper. The structure and writing would benefit from revisions to improve clarity and logical flow. In some sections, there are grammatical issues and information that seems less directly relevant to your central research questions. A careful edit could help streamline the narrative and strengthen the presentation.

Additionally, while the dataset is robust, the findings in their current form are somewhat limited in novelty. The discussion section, in particular, could be expanded to explore the implications of your results more deeply and to connect them more clearly with existing literature. It may also be helpful to consider additional contextual factors—such as socioeconomic status—that could be influencing variables like ethnic group or breakfast frequency, as these were not fully addressed in the current version.

Overall, I believe the manuscript has potential, but it would benefit from substantial revision before being suitable for publication. I encourage you to continue refining your work and hope these comments are helpful in guiding your next steps.

Reviewer #2: The article uses a solid methodology, explains the steps followed for the analysis, includes a statistically large population, and justifies the type of analysis using machine learning.

It meets ethical standards and has ethical approval.

The results are clear; despite the extensive information, they are presented in an understandable manner.

Throughout the document, there is good support with bibliographic references.

6. PLOS authors have the option to publish the peer review history of their article (what does this mean? ). If published, this will include your full peer review and any attached files.

**Do you want your identity to be public for this peer review?** For information about this choice, including consent withdrawal, please see our Privacy Policy .

Reviewer #1: No

Reviewer #2: **Yes: ** Patricia Philco-Lima

---

## [Author Response · Author response to Decision Letter 1]

15 Sep 2025

Additional Editor Comments:

Manuscript Title:

Machine Learning Analysis of Obesity Factors Across Weight Groups: Findings from the 2021 Youth Risk Behavior Surveillance System

General Assessment:

This manuscript addresses a timely and relevant public health issue—adolescent obesity—by applying supervised machine learning models to a large national dataset (YRBSS 2021). The incorporation of SHAP values to interpret feature importance is a methodological strength, offering transparency and clinical relevance. However, the manuscript requires substantial revision to clarify methodological choices, strengthen the scientific communication of results, and improve reproducibility and interpretability.

1. Title and Introduction

Concerns:

• The title does not reflect the use of SHAP or the study's focus on modifiable predictors, which are core to its novelty.

• The study aim appears too late in the introduction.

• The YRBSS dataset is not introduced early or described sufficiently.

• There is no justification provided for stratifying by weight categories.

Recommendations:

• Revise the title to reflect key methodological features. Suggested title:

“SHAP-Enhanced Machine Learning Identifies Modifiable Obesity Predictors Across Adolescent Weight Groups: A 2021 YRBSS Analysis.”

Response:

Thank you for this helpful suggestion. We have revised the title to highlight SHAP and modifiable predictors as advised.

• State the study objective clearly and early.

Response:

Thank you. We have moved the study objective earlier in the introduction to make it clear from the outset.

• Briefly describe the YRBSS dataset (e.g., scope, periodicity, population coverage).

Response:

Thank you. We have added a concise description of the YRBSS dataset earlier in the introduction.

• Justify the weight-group stratification using appropriate references (e.g., CDC percentile cutoffs for BMI-for-age).

Response:

Thank you. We now note in the introduction that participants were stratified by weight categories, and in the Methods section we provide justification using CDC BMI-for-age percentile cutoffs (BMIPCT).

2. Methods

Concerns:

• Unclear terminology (e.g., “two-step say technique”) may confuse readers.

• The definition of obesity and BMI percentile cutoffs is missing.

• Critical aspects of the machine learning pipeline are underreported:

No mention of hyperparameter tuning

No treatment of class imbalance

No description of cross-validation

•SHAP implementation details are insufficient.

• Reproducibility elements (software, random seed, code repository) are lacking.

Recommendations:

• Correct unclear terms or provide definitions.

Response:

Thank you. We have corrected the unclear term to “two-step imputation technique” and provided a brief explanation of how missing values were handled.

• Specify how BMI categories were defined and cite relevant standards (e.g., CDC, WHO).

Response:

Thank you. We have specified the definition of obesity and BMI percentile cutoffs in the Methods, citing CDC standards.

• Expand the ML pipeline description to include:

Hyperparameter optimization approach

Class imbalance techniques (e.g., SMOTE, weighting)

Validation strategy (e.g., k-fold CV, train-test split)

Response:

Thank you. We have revised the Methods to clarify our ML pipeline, including the random seed, software used, and SHAP implementation. We also explicitly state that we did not use hyperparameter optimisation, class imbalance techniques, or cross-validation, in order to prioritise transparency and reproducibility.

• Clearly explain how SHAP values were computed and aggregated (e.g., mean absolute values).

Response:

Thank you. We have clarified our SHAP implementation, specifying that SHAP values were aggregated using mean absolute values across all samples, and visualised with beeswarm plots.

• For reproducibility, list the software used, indicate random seed values, and share a code repository if available.

Response:

Thank you. We have added reproducibility details by specifying software versions, the random seed (set.seed = 3554), and a statement about code availability.

3. Results

Concerns:

• The results section includes redundant reporting (e.g., odds ratios repeated in text and tables).

• Overreporting of non-significant findings (e.g., soda, sleep) may distract from meaningful results.

• Performance metrics sensitive to class imbalance (e.g., F1-score, PR AUC) are missing.

• SHAP plots and nomograms are insufficiently labeled.

Recommendations:

• Streamline the text to highlight key, statistically significant findings.

Response:

Thank you. We have streamlined the Results section by highlighting only the key, statistically significant findings and moving non-significant results and redundant details to the supplementary materials.

• Move non-significant results to supplementary material.

Response:

Thank you. We have moved non-significant results to the supplementary materials and streamlined the Results section to emphasise only the key, statistically significant findings.

• Report additional evaluation metrics (e.g., F1-score: “F1 scores were 0.41 for Random Forest and 0.39 for XGBoost”).

Response:

Thank you. We have added imbalance-sensitive evaluation metrics, including weighted F1-scores (0.277 for Random Forest; 0.292 for XGBoost) and mean one-vs-rest AUC (0.575 and 0.571). We also reported class-level F1-scores to highlight the effect of class imbalance.

• Improve figure labeling:

Label SHAP axes (e.g., “SHAP value (impact on log-odds of obesity)”)

Clarify nomogram elements with legends or captions.

Response:

Thank you. We revised the figure labels and captions. SHAP plots now include axis labels and colour explanations, and the nomogram caption clarifies how points and predicted values are derived.

• State that top features were selected based on mean absolute SHAP values.

Response:

Thank you. We have clarified in the Results section that the top features were selected based on mean absolute SHAP values.

4. Discussion

Concerns:

• The discussion contains redundant commentary on interpretability.

• Terminology is inconsistent or unclear (e.g., “butter level,” “fresh BMI,” “quiet activities”).

• Overgeneralizations are made, particularly regarding dietary predictors.

• The narrative flow is disjointed.

• The limitations section is underdeveloped.

Recommendations:

• Consolidate discussion on SHAP and interpretability.

Response:

Thank you. We have consolidated the discussion on interpretability and SHAP into a single paragraph, reducing redundancy while emphasising its role in highlighting key predictors.

• Use consistent, scientific terminology (e.g., “butter level” → “milk intake,” “fresh BMI” → “BMI status”).

Response:

Thank you. We have revised inconsistent terminology, replacing unclear terms with consistent scientific expressions.

• Clarify overgeneralized findings (e.g., specify milk type/frequency).

Response:

Thank you. We have revised the discussion of dietary predictors to avoid overgeneralization, and clarified that the association with milk depends on both type (whole vs. low-fat) and frequency of intake.

• Improve discussion structure using subheadings (e.g., MVPA, screen time, substance use).

Response:

Thank you. We have reorganised the discussion with subheadings (e.g., MVPA, screen time, dietary predictors, substance use) to improve structure and readability.

• Expand the limitations section:

Cross-sectional design (no causal inference)

Self-reported data bias

Lack of objective activity measures (e.g., wearables)

Absence of longitudinal validation

Response:

Thank you. We have expanded the limitations section to address the cross-sectional design, reliance on self-reported data, lack of objective measures, and absence of longitudinal validation.

5. Statistical Analysis Clarity

Concern:

The authors mention the use of accuracy, sensitivity, specificity, and AUC to evaluate model performance but do not detail how these metrics were calculated or validated.

Recommendation:

• Briefly describe whether performance metrics were calculated on a holdout set or through cross-validation.

Response:

Thank you. We have clarified that all performance metrics (accuracy, sensitivity, specificity, AUC) were calculated on the held-out testing set.

• Indicate how classification thresholds were set.

Response:

Thank you. We have clarified that classification thresholds were set according to the default algorithm behaviour, assigning each case to the class with the highest predicted probability.

• Discuss the relevance of class imbalance metrics if applicable.

Response:

Thank you. We have added an explanation that weighted F1-score and mean one-vs-rest AUC are particularly relevant under class imbalance, as they provide a more balanced evaluation of model performance.

6. Minor Issues

• Proofread for minor typographical and grammatical inconsistencies.

Response:

Thank you. We have carefully proofread the manuscript to correct minor typographical and grammatical inconsistencies, ensuring consistent and clear language throughout.

• Standardize variable names and phrasing throughout the manuscript.

Response:

Thank you. We have standardised variable names and phrasing throughout the manuscript to ensure consistency.

Final Recommendation:

Major Revision

The manuscript presents a valuable application of machine learning to a pressing health issue, but critical revisions are necessary to ensure scientific clarity, methodological transparency, and interpretability. I encourage the authors to address the above comments comprehensively.

Reviewer #1: Thank you for the opportunity to review your manuscript. This study analyzes a large dataset, which is a valuable strength and holds great potential to contribute to the field. I appreciate the effort that went into collecting and analyzing the data.

That said, there are a few areas where the manuscript could be improved to better communicate your findings and enhance the overall impact of the paper. The structure and writing would benefit from revisions to improve clarity and logical flow. In some sections, there are grammatical issues and information that seems less directly relevant to your central research questions. A careful edit could help streamline the narrative and strengthen the presentation.

Additionally, while the dataset is robust, the findings in their current form are somewhat limited in novelty. The discussion section, in particular, could be expanded to explore the implications of your results more deeply and to connect them more clearly with existing literature. It may also be helpful to consider additional contextual factors—such as socioeconomic status—that could be influencing variables like ethnic group or breakfast frequency, as these were not fully addressed in the current version.

Overall, I believe the manuscript has potential, but it would benefit from substantial revision before being suitable for publication. I encourage you to continue refining your work and hope these comments are helpful in guiding your next steps.

---

## [Editor Report · Decision Letter 1]

17 Sep 2025

PONE-D-25-31368R1SHAP-Enhanced Machine Learning Identifies Modifiable Obesity Predictors Across Adolescent Weight Groups: A 2021 YRBSS AnalysisPLOS ONE

Dear Dr. Yan,

Thank you for submitting your manuscript to PLOS ONE. After careful consideration, we feel that it has merit but does not fully meet PLOS ONE’s publication criteria as it currently stands. Therefore, we invite you to submit a revised version of the manuscript that addresses the points raised during the review process.

We look forward to receiving your revised manuscript.

Kind regards,

Mohamed Ahmed Said, Ph.D.

Academic Editor

PLOS ONE

Journal Requirements:

Additional Editor Comments:

Summary of Manuscript:

The manuscript examines factors associated with overweight and obesity among adolescents using data from the YRBSS dataset. The study applies multivariable logistic regression to evaluate associations between demographic, behavioral, and lifestyle variables and overweight/obesity status. Additionally, the authors present a predictive machine learning analysis using Random Forest and XGBoost models.

Major Comments:

Regression Analysis:

The logistic regression analyses are clearly presented with appropriate tables (Tables 2 and 3).

Demographic and behavioral predictors (age, sex, race, MVPA, diet, sleep, and substance use) are analyzed correctly, and results are generally interpretable.

Machine Learning Component:

The manuscript presents Random Forest and XGBoost results for obesity prediction.

However, the editorial assessment indicates no hyperparameter tuning, no class imbalance handling, and no cross-validation were performed.

Reported performance metrics (accuracy ~0.66–0.67, very low positive predictive value) are not robust, limiting reliability and generalizability.

The current ML analysis is insufficient to support predictive claims.

Clarity and Structure:

Introduction and discussion sections have improved in clarity and flow.

Methodology reporting is transparent for the regression analyses, but the ML methodology requires major improvements or reframing.

Minor Comments:

Tables are well organized; ensure consistent formatting for all p-values and odds ratios.

References are comprehensive and up-to-date.

Required Actions for Authors:

Authors must address the critical limitations in the ML component. Two options are available:

Perform a full ML re-analysis:

Apply hyperparameter tuning (grid/random search).

Implement class imbalance strategies (SMOTE, class weighting).

Use robust validation (k-fold or nested cross-validation).

Update results, figures, and conclusions accordingly.

Reframe the manuscript as descriptive/exploratory analysis:

Focus on associations identified in regression analyses.

Remove or tone down all predictive claims.

Clearly state the absence of ML validation and tuning as a major limitation.

Once the manuscript is revised accordingly, it can be re-evaluated for potential acceptance.

---

## [Author Response · Author response to Decision Letter 2]

26 Sep 2025

PLOS ONE

Journal Requirements:

Additional Editor Comments:

Summary of Manuscript:

The manuscript examines factors associated with overweight and obesity among adolescents using data from the YRBSS dataset. The study applies multivariable logistic regression to evaluate associations between demographic, behavioral, and lifestyle variables and overweight/obesity status. Additionally, the authors present a predictive machine learning analysis using Random Forest and XGBoost models.

Major Comments:

Regression Analysis:

The logistic regression analyses are clearly presented with appropriate tables (Tables 2 and 3).

Response: Thank you!

Demographic and behavioral predictors (age, sex, race, MVPA, diet, sleep, and substance use) are analyzed correctly, and results are generally interpretable.

Response: Thank you!

Machine Learning Component:

The manuscript presents Random Forest and XGBoost results for obesity prediction.

Response: Thank you!

However, the editorial assessment indicates no hyperparameter tuning, no class imbalance handling, and no cross-validation were performed.

Response: Thank you and we addressed this concern by performing grid search for hyperparameter tuning, applying SMOTE with class weights for imbalance handling, and using repeated 5-fold cross-validation to ensure robust evaluation. Please see:

Reported performance metrics (accuracy ~0.66–0.67, very low positive predictive value) are not robust, limiting reliability and generalizability.

Response: Thank you and we have added up some analysis in the limitation section:

“Additionally, model performance was modest (accuracy ~0.66–0.67, low positive predictive value), reflecting reliance on self-reported behavioural data. Despite tuning and imbalance handling, predictive power remained limited; thus, the ML analyses were intended to highlight correlates rather than provide precise prediction.”

The current ML analysis is insufficient to support predictive claims.

Clarity and Structure:

Introduction and discussion sections have improved in clarity and flow.

Response: Thank you!

Methodology reporting is transparent for the regression analyses, but the ML methodology requires major improvements or reframing.

Response: Thank you. We have revised the Statistical Analysis section to clarify the framing of the machine learning methodology.

Minor Comments:

Tables are well organized; ensure consistent formatting for all p-values and odds ratios.

Response: Thank you, changed!

References are comprehensive and up-to-date.

Response: Thank you.

Required Actions for Authors:

Authors must address the critical limitations in the ML component. Two options are available:

Perform a full ML re-analysis:

Apply hyperparameter tuning (grid/random search).

Implement class imbalance strategies (SMOTE, class weighting).

Use robust validation (k-fold or nested cross-validation).

Update results, figures, and conclusions accordingly.

Reframe the manuscript as descriptive/exploratory analysis:

Focus on associations identified in regression analyses.

Remove or tone down all predictive claims.

Clearly state the absence of ML validation and tuning as a major limitation.

Once the manuscript is revised accordingly, it can be re-evaluated for potential acceptance.

Response: We appreciate this constructive suggestion. In line with the first option, we have conducted a full re-analysis of the ML component, including hyperparameter tuning (grid search with cross-validation), class imbalance handling (SMOTE with class weighting), and repeated 5-fold cross-validation. Results, figures, and conclusions have been updated accordingly, while the ML analyses are framed as complementary to regression findings.

---

## [Editor Report · Decision Letter 2]

30 Sep 2025

SHAP-Enhanced Machine Learning Identifies Modifiable Obesity Predictors Across Adolescent Weight Groups: A 2021 YRBSS Analysis

PONE-D-25-31368R2

Dear Dr. Jin Yan,

We’re pleased to inform you that your manuscript has been judged scientifically suitable for publication and will be formally accepted for publication once it meets all outstanding technical requirements.

Kind regards,

Mohamed Ahmed Said, Ph.D.

Academic Editor

PLOS ONE
---

## [Editor Report · Acceptance letter]

PONE-D-25-31368R2

PLOS ONE

Dear Dr. Yan,

I'm pleased to inform you that your manuscript has been deemed suitable for publication in PLOS ONE. Congratulations! Your manuscript is now being handed over to our production team.

Kind regards,

on behalf of

Dr. Mohamed Ahmed Said

Academic Editor

PLOS ONE